# Burkitt Lymphoma Incidence in Five Continents

**Sam M. Mbulaiteye * and Susan S. Devesa**

Infections and Immunoepidemiology Branch, Division of Cancer Epidemiology and Genetics, National Cancer Institute, Bethesda, MD 20892, USA; devesas@nci.nih.gov

**\*** Correspondence: mbulaits@mail.nih.gov; Tel.: +1-240-276-7108

**Abstract:** Burkitt lymphoma (BL) is a rare non-Hodgkin lymphoma first described in 1958 by Denis Burkitt in African children. BL occurs as three types, endemic, which occurs in Africa and is causally attributed to Epstein-Barr virus and *P falciparum* infections; sporadic, which occurs in temperate areas, but the cause is obscure; and immunodeficiency-type, which is associated with immunosuppression. All BL cases carry *IG::MYC* chromosomal translocations, which are necessary but insufficient to cause BL. We report a comprehensive study of the geographic, sex, and age-specific patterns of BL among 15,122 cases from Cancer Incidence in Five Continents Volume XI for 2008–2012 and the African Cancer Registry Network for 2018. Age-standardized BL rates were high (>4 cases per million people) in Uganda in Africa, and Switzerland and Estonia in Europe. Rates were intermediate (2–3.9) in the remaining countries in Europe, North America, and Oceania, and low (<2) in Asia. Rates in India were 1/20th those in Uganda. BL rates varied within and between regions, without showing a threshold to define BL as endemic or sporadic. BL rates were twice as high among males as females and showed a bimodal age pattern with pediatric and elderly peaks in all regions. Multi-regional transdisciplinary research is needed to elucidate the epidemiological patterns of BL.

**Keywords:** Burkitt lymphoma; epidemiology; *Plasmodium falciparum*; Epstein Barr virus; registry studies; multimodal cancer; non-Hodgkin lymphoma; HIV/AIDS

## 1. Introduction

Burkitt lymphoma (BL) is an aggressive B-cell non-Hodgkin lymphoma of germinal center B cells [1], first described as a jaw sarcoma in Ugandan children by Denis Burkitt in 1958 [2]. The pathology of BL was defined as a diffuse infiltration of medium-sized lymphoid cells, with pale staining macrophages giving a "starry-sky" appearance under the microscope [3,4]. This pathology definition enabled the epidemiology and clinical behavior of BL cases to be studied worldwide [5–8], at different anatomic sites [4,9,10], and from periods predating Burkitt's report [11]. The World Health Organization (WHO) introduced the eponym "Burkitt tumor" to standardize the classification of BL cases worldwide and to recognize Burkitt's seminal contribution [12].

The early study of BL revealed a comparatively high incidence of BL cases in warm, wet, low-lying areas, particularly in Africa, and a low incidence of BL cases in temperate or arid areas [13]. This simple distribution was used to classify BL cases as endemic or sporadic, and provided a useful insight about possible causal factors, notably that BL was caused by a virus [14,15] or parasite, such as malaria, vectored by insects in Africa [16]. Discovery of Epstein-Barr virus (EBV), by electron microscopy in tumor samples obtained from an African child [17], appeared to confirm the virus hypothesis. EBV DNA was detected in ~95% of endemic BL cases, consistent with this hypothesis. However, EBV DNA was detected in only 10–20% of sporadic BL cases, suggesting that EBV was not necessary for development of BL. EBV also did not satisfy the "insect-vector transmission mode" of the hypothesis because it was transmitted by saliva [18,19] and was equally prevalent in

areas with endemic and sporadic BL. The insect-vector hypothesis was satisfied by holo-endemic *Plasmodium falciparum*, which is transmitted by female *Anopheline* mosquitoes and is co-endemic with BL [20,21]. Thus, the hypothesis was modified to suggest that interactions between EBV and *P falciparum* infection promote BL risk, with *P falciparum* promoting B-cell proliferation through chronic stimulation of the reticulo-endothelial system [21] and increasing EBV burden [22]. The hypothesis that EBV and *P falciparum* cooperate in BL development is one of the earliest examples of polymicrobial etiology of cancer [23]. Subsequently, the identification of BL in previously healthy young homosexual men in the US in the early 1980s [24] led to recognition of the acquired immunodeficiency syndrome (AIDS) [25] and focused attention on immunosuppression as a co-factor in BL etiology. For a short time, BL was used as a sentinel condition in the surveillance of AIDS as one of the AIDS-defining cancers [26].

Molecular studies showing consistent detection of *IG::MYC* translocations [27,28] confirmed common pathology of BL reported in diverse settings. The translocation is believed to be an early, possibly initiating event, in all forms of BL. Clinical studies, demonstrating similar responses to chemotherapy for endemic cases in Africa [29–31] and sporadic cases in the US and elsewhere [32,33], reinforced the notion that BL cases diagnosed in different settings shared common pathology. Research on BL marks a watershed moment in cancer research, which gave birth to new specialties, including discovery of tumor viruses, elucidation of biology, re-focusing research on chemotherapy to combination regimens with intent to cure cancer, and increased interest in global oncology [34]. The excitement triggered by BL prompted Joseph Burchenal to declare in 1966 [35] that geographic studies of BL would reveal new principles in cancer research. These new principles include six human tumor viruses that have been discovered [36] and now are linked to ~1 million cancers worldwide (about one-sixth of the cancer burden) [37] that are prime targets for cancer prevention.

However, our understanding of the worldwide epidemiology of BL is incomplete because current research of BL does not have a global coverage [15,38]. Our understanding is also limited by the pathology definition established six decades ago, which did not codify a gold standard [39,40]. For example, the pathology definition recognized variant pathology features, such as differences in cell size, shape, degree of differentiation, amount of stroma and number of histocytes. BL cases with variant pathology have been described as atypical BL or Burkitt-like [41], and were classified and coded with BL until 2008 [42,43]. Thus, it is difficult to compare BL data from single countries [44–48] or in limited geographical regions [15,49–53] because the variant BL cases may be treated differently and the results are difficult to generalize. For example, the incidence rate of BL in the US increased steeply (~7% per year) between 1973 and 2005 and exhibited pediatric, adult, and elderly age-specific peaks [46,47]. While the adult patterns could be attributed to the underlying HIV epidemic in the US [47], this explanation was difficult to reconcile with the lack of corresponding increases in BL in Africa, one of the regions with many cases of AIDS [54], or in India and China, which also have a substantial burden of HIV [55–57]. An attempt to resolve this issue by analyzing BL data for cases diagnosed in four continents between 1963–2002 [58] confirmed the temporal patterns of BL and the multiple age-specific peaks but not the correlation between BL and regional HIV prevalence. Data in this study were sparse or not available for sub-Saharan Africa, North Africa, the Middle East, and the Caribbean.

We conducted an epidemiological analysis of 15, 122 BL cases diagnosed worldwide using data reported to the International Agency for Research on Cancer (IARC) Cancer Incidence in Five Continents Volume XI (CI5(XI)) for 2008–2012 [59] and estimated by the African Cancer Registry Network (AFRCN) for 2018 [50].

## 2. Materials and Methods

The CI5(XI) data [59] were compiled from cancer registries in 65 countries (details in Supplementary Table S1). The cancer coverage was sub-national in most countries, except

for nineteen that have national coverage (36 countries had a single registry and twenty-nine had multiple registries). The CI5(XI) report represents about 15% of the worldwide population. The lowest coverage is in Africa (1%), Central and South America (8%), and Asia (7%), while near half or higher are in Europe (46%), Oceania (77%), and the highest in North America (98%).

Cancer data were available for the 5-year period 2008-2012 in most countries, but only for shorter periods (four or three years, Supplementary Table S1) in a few. BL was defined as cases with International Classification of Diseases, 10th edition (ICD10) code = C83.7 [60]. BL data were downloaded by registry as case counts and person-years by gender and 5-year age group (Supplementary Table S2). Burkitt cell leukemia (BCL), which is coded as C91.8 (mature B-cell leukemia, Burkitt type) was not coded with BL in CI5(XI). Data are presented by country. Data from multiple registries in a country were aggregated to calculate country-specific estimates. US data were available for white and black people as well as overall. To minimize the impact of sparse data on the rates, we restricted analyses to countries with a minimum of total cases. Specifically, 30 total cases for geographic analyses of overall rates (38 countries), 60 total cases for sex-specific analyses (28 countries), and 100 total cases for age-specific analyses (18 countries), referred to as the 30/60/100 criteria. Because cancer data cover only 1% of people in Africa, we supplemented the African data with estimates published by AFRCN for 2018 [60]. AFRCN estimated BL rates by calculating the proportions of NHL (ICD10 codes C82–86, C96) in GLOBOCAN 2018 within 5 broad sex-specific age groups that were attributed to BL [50]. The calculated proportions were applied to the estimated number of NHL cases (by sex and age) in GLOBOCAN 2018 for the African countries (Supplementary Table S3). Countries that have no registries were included by using the mean of the proportions (within age–sex groups) from the nearest neighboring countries with cancer registries. We filtered these data using 30/60/100 total case criteria to include data for 27 countries for the geographic patterns, 16 countries for sex-specific patterns, and 10 countries for age-specific patterns (Supplementary Table S3).

### 2.1. Age-Standardized Incidence Rates

Age-standardized incidence rates (ASR) of BL per million person-years were calculated by country using the CI5(XI) data, adjusted to the World Standard Population of Segi and Doll [61]. The available AFRCN data were age-adjusted using the same standard. The overall BL ASRs were plotted on the world map to discern geographic patterns. The sex-specific rates were plotted on horizontal bar charts, sorted in descending order by the rates among males by region and within region to discern regional patterns. Male-to-female incidence rate ratios were calculated and tabulated.

### 2.2. Age-Specific Incidence Rates

Age-adjusted age-specific BL rates were calculated for 5 age groups (0–14, 15–34, 35–54, 55–74, 75+). Our goal was to confirm or refute the multimodal patterns hypothesized based on previous reports in the US [46,47,62] and data from four continents but not Africa [58]. Rates based on ≥10 cases were plotted on a log scale on the y-axis and age intervals on the x-axis on a linear interval scale (Supplementary Table S4). We compared age-specific incidence rate ratios (IRRs) for BL in different countries versus rates among US white people, which we selected as the referent because it was the largest group in the dataset. The IRR results were plotted as horizontal bar charts to discern patterns.

### 3. Results

#### 3.1. Worldwide Geographic Patterns of BL

We downloaded data for 11,743 BL cases from the CI5(XI) report during 2008–2012 (Supplementary Table S2) and retained for geographic analysis 11,446 BL cases from 38 countries with ≥30 total BL cases (1 in Africa, 5 in Central or South America, the US and

Canada in North America, 9 in Asia, 19 in Europe, and 2 in Oceania). Five countries accounted for 67.9% of BL cases (Table 1): US (47.2%) in North America; United Kingdom (8.3%), and Germany (5.9%) in Europe; South Korea (3.3%) in Asia, and Australia in Oceania (3.2%).

Figure 1a and Table 1 show the geographic distribution of BL worldwide. Using CI5(XI) data, we observed a 20-fold range between Uganda (the country with the highest BL ASR) and China (the country with the lowest BL rate). In addition, BL rates varied 2–8-fold within and between regions. To simplify reporting, the rates were categorized descriptively and arbitrarily as "high", defined as ≥ 4 cases per million person-years, "intermediate", defined as 2–3.9 cases per million, and "low", defined as <2 cases per million. All regions included countries with high, intermediate, or low BL rates. In CI5(XI) data, BL rates were high in Uganda (9.28 per million), Switzerland, and Estonia, with 4.12 and 5.67 cases per million, respectively. BL rates were intermediate in the US in North America and in Central/South America; in Israel, Saudi Arabia, and Turkey in Asia; and in most countries in Europe. BL rates were low in most countries in Asia and the Czech Republic, Belarus, Ukraine, Poland, and the Russian Federation in Europe. Worldwide, BL rates were lowest in China (0.45) and India (0.59). The BL incidence rate ratio (IRR), relative to the rate among white people in the US, was about three for Uganda and less than 0.2 for China and India.

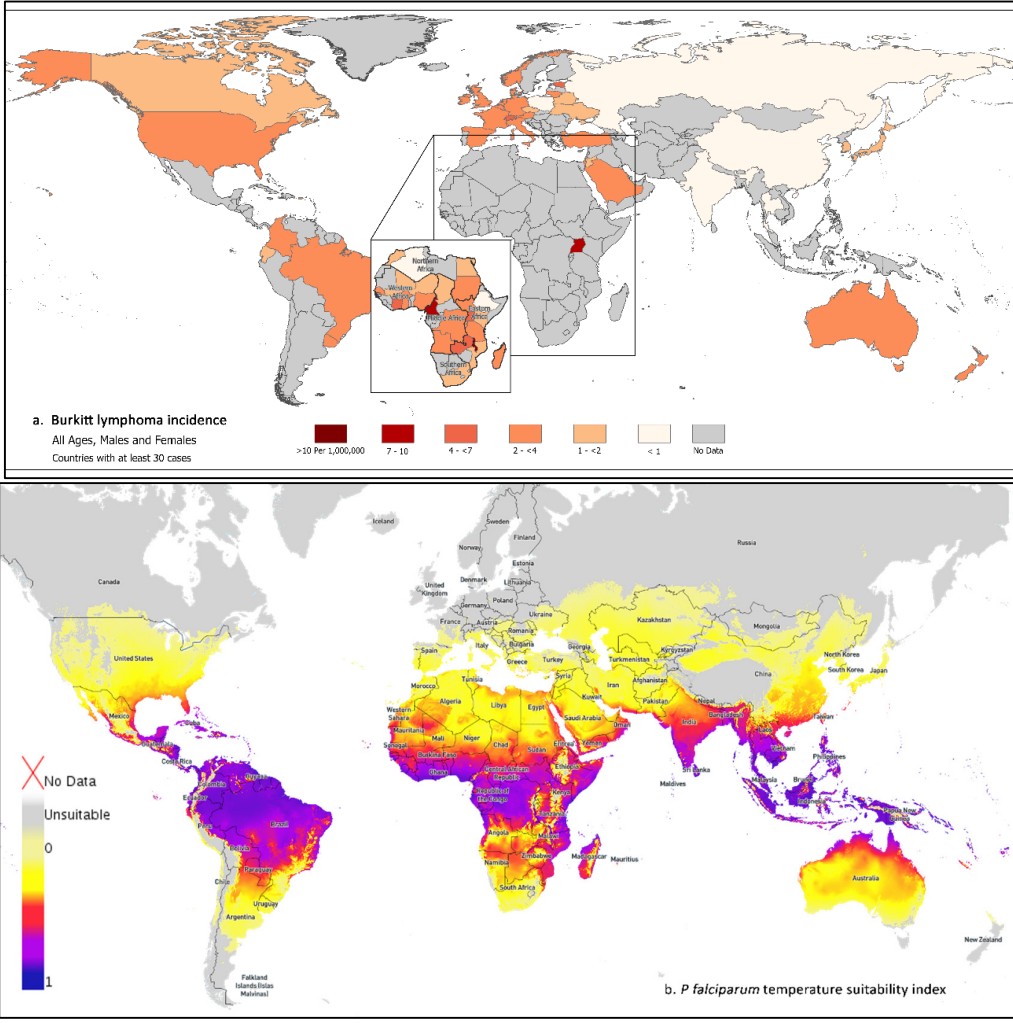

**Figure 1.** (**a**) Age-standardized Burkitt lymphoma rates per million person-years (world standard) for males and females combined in 38 countries with at least 30 cases during 2008–2012, using data

from CI5(XI). Inset shows data for Africa in 27 countries with at least 30 cases during 2018 from the AFRCN. (**b**) World map showing *P falciparum* temperature suitability index for *P falciparum* transmission 2010 (data from the Malaria Atlas Project: https://malariaatlas.org/explorer/#/; accessed 9 June 2022).

**Table 1.** Burkitt lymphoma incidence by region and country for both sexes combined, sorted by rate within region, based on data from Cancer Incidence in Five Continents Volume XI [CI5(XI)] 2008–2012 and the African Cancer Registry Network (AFRCN) 2018.

| Region | Country [1] | Total Cases [2] | Total Rate [3] | Incidence Rate Ratio [4] |
|---|---|---|---|---|
| **International CI5(XI) data for 2008-2012:** | | | | |
| **Africa** | | | | |
| | Uganda | 124 | 9.28 | 2.91 |
| **Central/South America** | | | | |
| | Colombia | 60 | 3.19 | 1.00 |
| | Puerto Rico | 60 | 3.13 | 0.98 |
| | Uruguay | 48 | 2.82 | 0.88 |
| | Brazil | 41 | 2.23 | 0.70 |
| | Ecuador | 33 | 1.15 | 0.36 |
| **North America** | | | | |
| | US white people (referent) | 4482 | 3.19 | 1.00 |
| | United States (US) | 5405 | 3.09 | 0.97 |
| | US black people | 611 | 2.68 | 0.84 |
| | Canada | 194 | 1.47 | 0.46 |
| **Asia** | | | | |
| | Israel | 143 | 3.77 | 1.18 |
| | Saudi Arabia: Saudi | 54 | 2.41 | 0.76 |
| | Turkey | 115 | 2.30 | 0.72 |
| | Republic of Korea | 376 | 1.72 | 0.54 |
| | Japan | 276 | 1.55 | 0.49 |
| | Jordan: Jordanians | 39 | 1.17 | 0.37 |
| | Thailand | 44 | 0.88 | 0.28 |
| | India | 128 | 0.59 | 0.18 |
| | China | 120 | 0.45 | 0.14 |
| **Europe** | | | | |
| | Estonia | 35 | 5.67 | 1.78 |
| | Switzerland | 96 | 4.12 | 1.29 |
| | Belgium | 199 | 3.72 | 1.16 |
| | Norway | 90 | 3.49 | 1.10 |
| | Spain | 153 | 3.34 | 1.05 |
| | Italy | 347 | 3.23 | 1.01 |
| | The Netherlands | 265 | 3.16 | 0.99 |
| | France | 172 | 3.13 | 0.98 |
| | Lithuania | 37 | 3.00 | 0.94 |
| | Denmark | 77 | 2.92 | 0.91 |
| | United Kingdom | 950 | 2.68 | 0.84 |
| | Ireland | 67 | 2.64 | 0.83 |
| | Austria | 99 | 2.44 | 0.77 |
| | Germany | 676 | 2.43 | 0.76 |
| | Czech Republic | 91 | 1.97 | 0.62 |
| | Belarus | 70 | 1.96 | 0.61 |

| | | | |
|---|---|---|---|
| Ukraine | 252 | 1.52 | 0.48 |
| Poland | 37 | 0.88 | 0.27 |
| Russian Federation | 30 | 0.72 | 0.22 |
| **Oceania** | | | |
| New Zealand | 80 | 3.21 | 1.01 |
| Australia | 363 | 3.12 | 0.98 |
| | | | |
| **African AFRCN data for 2018:** | | | |
| **Eastern Africa** | | | |
| Malawi | 521 | 19.3 | 6.05 |
| Uganda | 307 | 4.8 | 1.50 |
| Zambia | 105 | 4.2 | 1.32 |
| Rwanda | 42 | 3.5 | 1.10 |
| Burundi | 37 | 3.4 | 1.07 |
| South Sudan | 39 | 2.5 | 0.78 |
| Tanzania | 171 | 2.2 | 0.69 |
| Madagascar | 71 | 2.1 | 0.66 |
| Kenya | 102 | 1.7 | 0.53 |
| Mozambique | 72 | 1.7 | 0.53 |
| Ethiopia | 56 | 0.4 | 0.13 |
| **Middle Africa** | | | |
| Cameroon | 251 | 8.0 | 2.51 |
| Congo, Democratic People Republic of | 261 | 2.9 | 0.91 |
| Angola | 86 | 2.1 | 0.66 |
| Chad | 38 | 1.7 | 0.53 |
| **Northern Africa** | | | |
| Sudan | 90 | 2.0 | 0.63 |
| Egypt | 178 | 1.7 | 0.53 |
| Morocco | 55 | 1.7 | 0.53 |
| Algeria | 37 | 0.9 | 0.28 |
| **Southern Africa** | | | |
| South Africa | 94 | 1.6 | 0.50 |
| **Western Africa** | | | |
| Cote d'Ivoire | 138 | 4.6 | 1.44 |
| Nigeria | 647 | 2.8 | 0.88 |
| Ghana | 86 | 2.4 | 0.75 |
| Senegal | 48 | 2.4 | 0.75 |
| Burkina Faso | 58 | 2.2 | 0.69 |
| Mali | 41 | 1.4 | 0.44 |
| Niger | 45 | 1.2 | 0.38 |

[1] In the CI5(XI) data, Uganda and Saudi Arabia each had 1 sub-national registry, Colombia and the Russian Federation had 4, Ecuador and Poland had 5, Brazil had 6, Thailand had 7, Turkey had 8, Japan, Switzerland, and Germany had 9, Canada had 12, Spain had 14, France had 15, India had 16, China and Italy had 36. [2] Includes countries with at least 30 total cases. [3] Rates per million person-years, age-adjusted using the 1960 Segi world standard. [4] Incidence rate ratio compared to rates of US white people.

The AFRCN dataset included 3899 estimated BL cases in 54 African countries in 2018 (Supplementary Table S3) and 3676 BL cases from 27 countries with ≥30 cases (Table 1, Figure 1, inset map for Africa). Five countries accounted for 54.0% of the cases: Nigeria

(17.6%), Malawi (14.2%), Uganda (8.4%), Democratic Republic of Congo (7.1%), and Cameroon (6.8%). While the CI5(XI) data from Uganda only appeared to suggest a threshold in the BL rates that could be used to classify cases in Africa (based on Uganda only) as endemic and those diagnosed elsewhere as sporadic BL, this apparent threshold was an artifact of sparse data. No threshold was evident when we used the AFRCN data, which showed a mosaic pattern with various countries in Africa having high, intermediate, or low BL rates. The highest and lowest BL rates were in East Africa where the Malawi rate of 19.3 was 48 times that of 0.4 in Ethiopia. BL rates were high (≥4) in five countries (Malawi, Cameroon, Uganda, Cote d'Ivoire, and Zambia) in low-lying regions in the Congo-Nile basin [15]. This pattern also corresponds to regions that are most prone to *P falciparum* transmission (Figure 1b). BL rates were intermediate (2.0–3.9) in Eastern Africa in Rwanda, Burundi, South Sudan, Tanzania, and Madagascar; in Middle Africa in the Democratic Republic of Congo and Angola; in Western Africa in Nigeria, Ghana, Senegal, and Burkina Faso; and in Northern Africa in Sudan. BL rates were low (<2.0) in Eastern Africa in Kenya, Mozambique, and Ethiopia; in Middle Africa in Chad; in Northern Africa in Egypt, Morocco, and Algeria; in Western Africa in Niger and Mali; and in South Africa in Southern Africa.

Table 2 and Figure 2a,b show the sex-specific BL rates in the CI5(XI) and AFRCN data for countries with at least 60 total cases. BL rates were higher in males than females in all countries. The male-to-female ratio exceeded 2.00 in most countries and 4.00 in seven countries. By far, BL rates were highest among males and females in Uganda and Malawi, followed by Cameroon. The lowest rates among both males and females (<1.0) were in India and China. The sex-specific rates showed within and between regional variation.

**Table 2.** Burkitt lymphoma incidence by region and country by sex, sorted by male rate within region, based on data from Cancer Incidence in Five Continents Volume XI [CI5(XI)] 2008–2012 and the African Cancer Registry Network (AFRCN) 2018.

| Region | Country [1] | MALES | | FEMALES | | |
| | | Cases [2] | Rate [3] | Cases [2] | Rate [3] | Male-to-Female Rate Ratio [4] |
|---|---|---|---|---|---|---|
| **International CI5(XI) data for 2008-2012:** | | | | | | |
| **Africa** | | | | | | |
| | Uganda | 71 | 11.74 | 53 | 7.05 | 1.67 |
| **Central/South America** | | | | | | |
| | Puerto Rico | 49 | 5.32 | 11 | 1.03 | 5.14 |
| | Colombia | 47 | 5.10 | 13 | 1.38 | 3.71 |
| **North America** | | | | | | |
| | US white people | 3344 | 4.94 | 1138 | 1.45 | 3.41 |
| | United States (US) | 3997 | 4.75 | 1408 | 1.46 | 3.25 |
| | US black people | 433 | 4.04 | 178 | 1.45 | 2.79 |
| | Canada | 151 | 2.35 | 43 | 0.58 | 4.02 |
| **Asia** | | | | | | |
| | Israel | 99 | 5.36 | 44 | 2.17 | 2.47 |
| | Turkey | 87 | 3.42 | 28 | 1.12 | 3.07 |
| | Republic of Korea | 289 | 2.71 | 87 | 0.70 | 3.86 |
| | Japan | 205 | 2.54 | 71 | 0.57 | 4.45 |
| | India | 80 | 0.70 | 48 | 0.47 | 1.47 |
| | China | 77 | 0.55 | 43 | 0.34 | 1.63 |
| **Europe** | | | | | | |
| | Switzerland | 71 | 6.63 | 25 | 1.54 | 4.31 |
| | Norway | 69 | 5.53 | 21 | 1.37 | 4.03 |
| | Belgium | 129 | 5.16 | 70 | 2.26 | 2.28 |

| | | | | | |
|---|---|---|---|---|---|
| The Netherlands | 202 | 5.02 | 63 | 1.25 | 4.02 |
| Italy | 255 | 5.02 | 92 | 1.38 | 3.65 |
| Spain | 108 | 4.89 | 45 | 1.72 | 2.84 |
| France | 122 | 4.70 | 50 | 1.56 | 3.02 |
| Denmark | 57 | 4.56 | 20 | 1.23 | 3.72 |
| United Kingdom | 695 | 4.12 | 255 | 1.25 | 3.30 |
| Austria | 76 | 4.10 | 23 | 0.75 | 5.47 |
| Ireland | 48 | 3.98 | 19 | 1.29 | 3.09 |
| Germany | 485 | 3.75 | 191 | 1.09 | 3.43 |
| Czech Republic | 66 | 3.06 | 25 | 0.84 | 3.65 |
| Belarus | 47 | 3.00 | 23 | 0.87 | 3.47 |
| Ukraine | 166 | 2.22 | 86 | 0.78 | 2.84 |
| **Oceania** | | | | | |
| New Zealand | 60 | 4.94 | 20 | 1.52 | 3.25 |
| Australia | 279 | 4.90 | 84 | 1.33 | 3.70 |
| **African AFRCN data for 2018:** | | | | | |
| **Eastern Africa** | | | | | |
| Malawi | 405 | 29.7 | 116 | 8.7 | 3.41 |
| Zambia | 84 | 6.6 | 21 | 1.7 | 3.88 |
| Uganda | 201 | 5.9 | 106 | 3.7 | 1.59 |
| Tanzania | 114 | 2.8 | 57 | 1.5 | 1.87 |
| Madagascar | 39 | 2.3 | 32 | 1.9 | 1.21 |
| Kenya | 67 | 2.2 | 35 | 1.2 | 1.83 |
| Mozambique | 36 | 1.7 | 36 | 1.7 | 1.00 |
| **Middle Africa** | | | | | |
| Cameroon | 159 | 10.4 | 92 | 5.6 | 1.86 |
| Congo, Democratic People Republic of | 179 | 4.3 | 82 | 1.5 | 2.87 |
| Angola | 68 | 3.2 | 18 | 0.9 | 3.56 |
| **Northern Africa** | | | | | |
| Sudan | 63 | 2.7 | 27 | 1.3 | 2.08 |
| Egypt | 137 | 2.5 | 41 | 0.8 | 3.13 |
| **Southern Africa** | | | | | |
| South Africa | 52 | 1.9 | 42 | 1.4 | 1.36 |
| **Western Africa** | | | | | |
| Cote d'Ivoire | 76 | 5.2 | 62 | 3.9 | 1.33 |
| Ghana | 68 | 3.7 | 18 | 1.0 | 3.70 |
| Nigeria | 379 | 3.0 | 268 | 2.5 | 1.20 |

[1] In the CI5(XI) data, Uganda had 1 sub-national registry, Colombia had 4, Turkey had 8, Japan, Switzerland, and Germany had 9, Canada had 12, Spain had 14, France had 15, India had 16, China and Italy had 36. [2] Includes countries with at least 60 total cases. [3] Rates per million person-years, age-adjusted using the 1960 Segi world standard. [4] Incidence Rate Ratio, Male Rate relative to Female Rate.

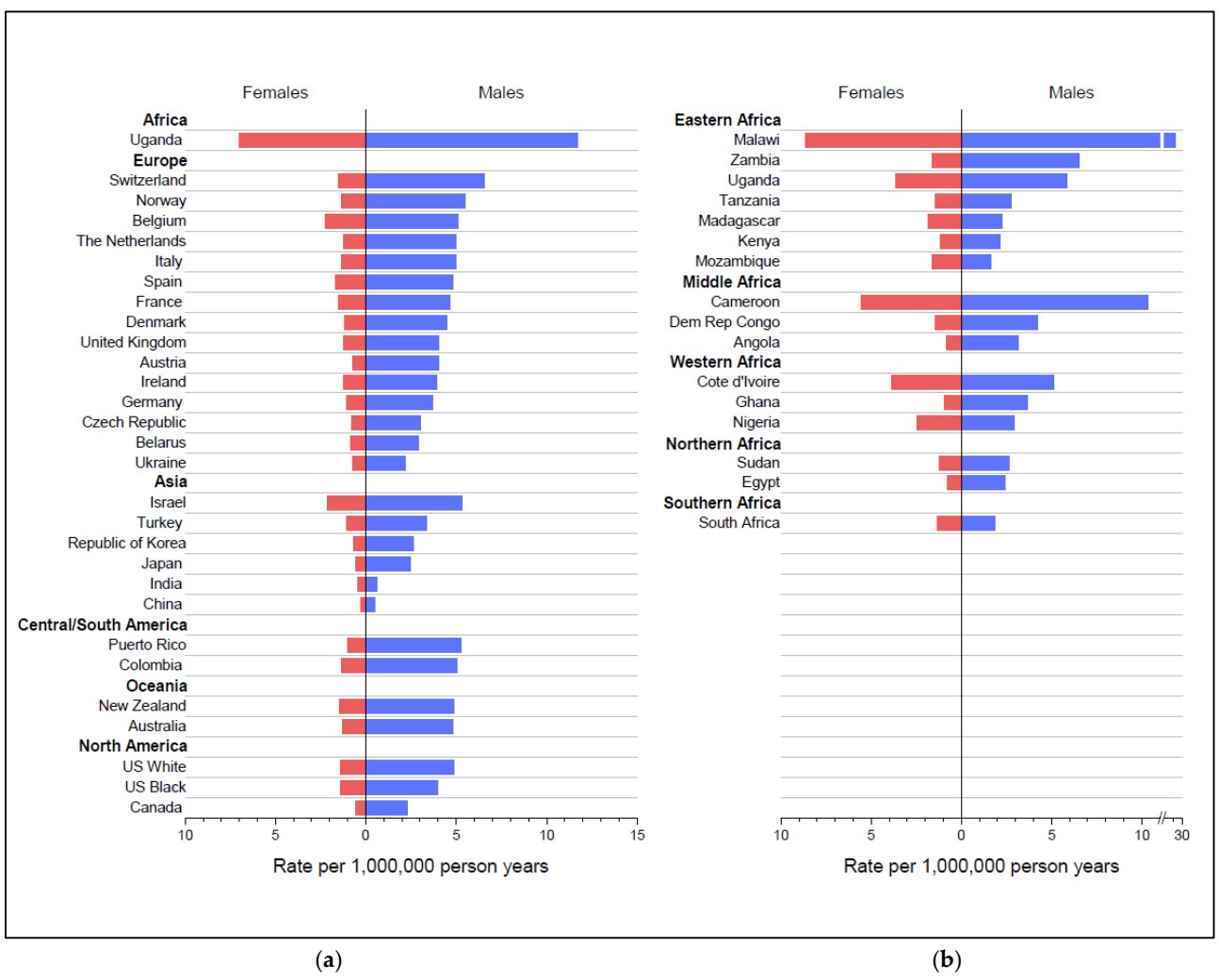

(**a**)          (**b**)

**Figure 2.** Age-standardized Burkitt lymphoma rates per million person-years (world) for males and females for countries with at least 60 cases total: (**a**) for 28 countries included in CI5(XI) and (**b**) for 16 countries included in the AFRCN data. The x-axis scale for Malawi is broken to accommodate the higher male BL rate in that country.

The proportion of BL cases aged 0–14 years was ~13% in the US, ~50% in Israel and Turkey, ~62% of cases in the Democratic Republic of Congo, 75% in Uganda, and 100% in Malawi and Zambia (Supplementary Table S4). BL incidence exhibited a bimodal pattern, with an early peak in the pediatric (0–14 years) age group and a second peak in the adult/elderly (55+) age groups (Figure 3). BL rates were lower for ages 15–54 years than the pediatric and elderly rates, representing the trough of a U-shape age-specific incidence curve. This shape of the age-specific BL rates was observed in all regions, except among US white people, US black people, and Australia where rates among adults were higher than the pediatric rates but lower than the elderly rates. The BL rates in one age group were correlated with rates in the other age groups, suggesting that when BL rates were intermediate in one age group, they were intermediate in the other age groups as well.

While country-specific data were sparse at ages ≥35 years, making it difficult to discern patterns at older ages for most African countries, bimodal patterns were observed in data aggregated by region. The exception was South Africa whose pattern resembled that in US white and black people, Australia, and the UK.

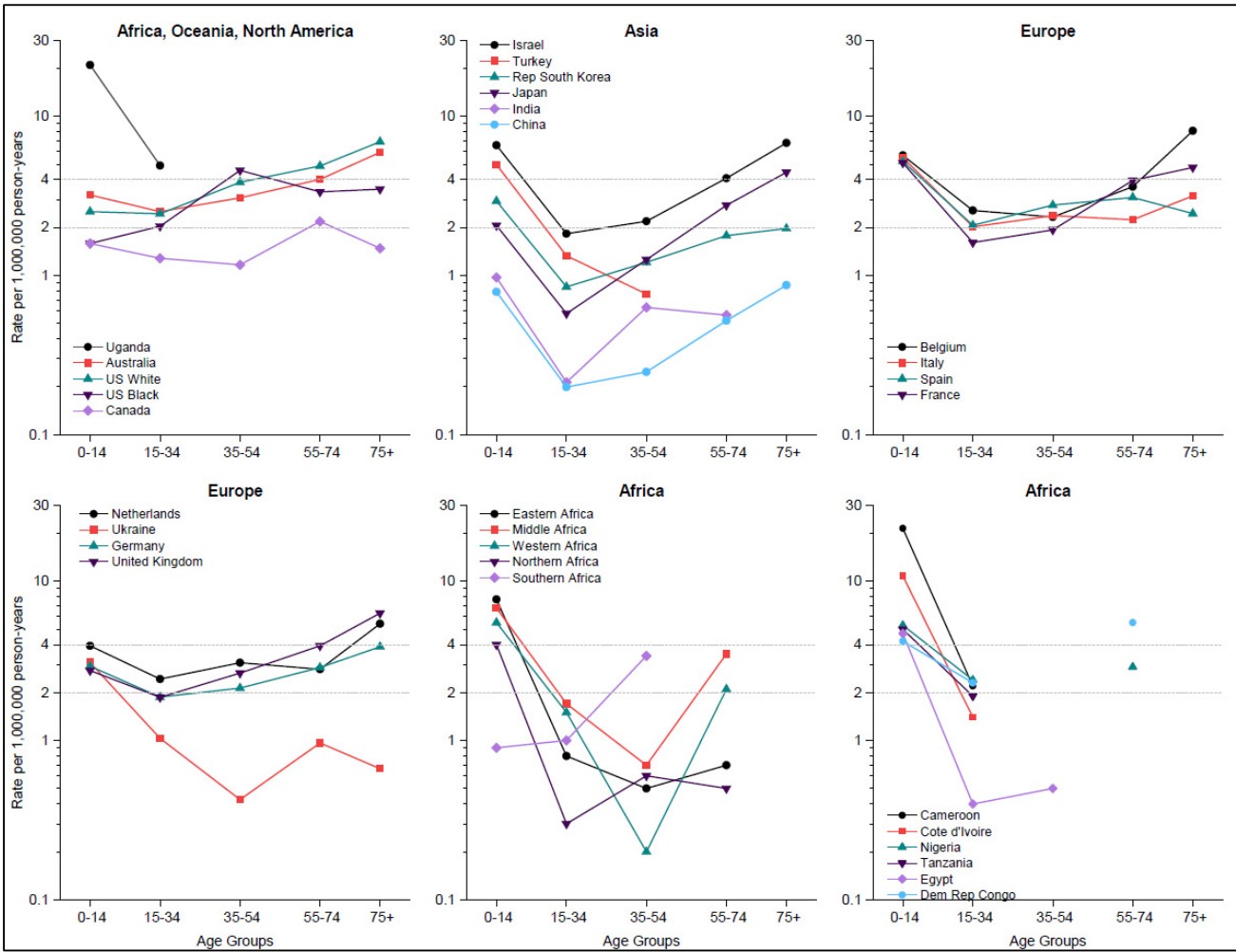

**Figure 3.** Burkitt lymphoma incidence rates for five age groups for countries with ≥100 total cases: Rates for 18 countries in five continents are based on CI5(XI) data for 2008–2012, and rates for Africa are supplemented with data reported to the AFRCN for 2018 by region and separately for six countries with cases in both 0–14 and 15–34 age groups. Rates based on <10 cases are not shown. Horizontal lines for BL rates = 2 and 4 cases per million divide the graph space into three areas corresponding to low, intermediate, and high BL rates (see results).

### 3.2. Age-Specific Incidence Rates Relative to BL Rates in US White Individuals

Figure 4 shows geographic variation in age-specific IRR as compared to US white people rates as the referent group (Supplementary Table S4). Among children aged 0–14 years, the IRRs ranged between 2.0–3.0 in Israel, Belgium, Italy, Spain, and France, and exceeded 8.0 in Uganda, in contrast to IRRs <0.5 in India and China. Among adults aged ≥55 years, IRRs were <0.5 in Canada, Korea, and Italy, and ≤0.2 in India, China, and Ukraine. Among those aged ≥75, IRRs were close to 1.0 only in Israel, Belgium, and the United Kingdom. Among US black individuals, all the IRRs were <0.85 except 35–54 where it was 1.2. Rates among US white adults aged 35 and older were the highest among all groups except US black people aged 35–54, those aged 75+ in Belgium, and those aged 55–74 in the Democratic Republic of Congo. Among the 10 countries in Africa, the IRRs for those aged 0–14 ranged from 1.6 to 2.0 in four countries, from 2.1 to 5.6 in four countries, to 8.5 in Cameroon and 24.6 in Malawi.

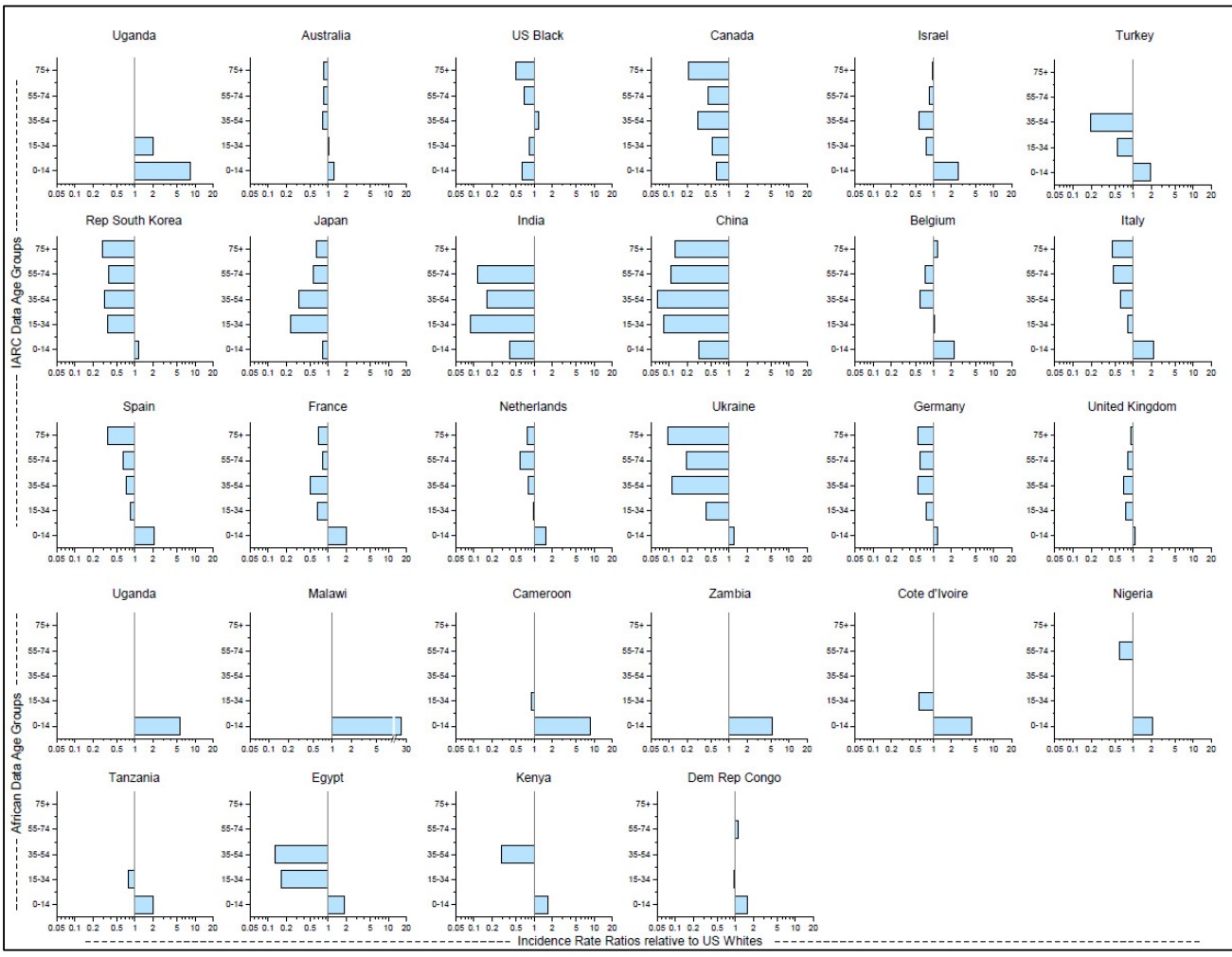

**Figure 4.** Age-specific Incidence Rate Ratios (IRR) of Burkitt lymphoma with the rate for US white individuals as the reference group. IRRs are plotted as horizontal bars along on the x-axis and the age groups on the y-axis. The first three rows are IRR using CI5(XI) data, while the bottom two rows use AFRCN data.

## 4. Discussion

Our report presents the most complete descriptive analysis of population profiles of BL, based on 15,122 BL cases diagnosed in 64 counties in five continents, including 26 from Africa. The results reveal three key insights. First, that there is no quantitative threshold for classifying all BL cases from Africa as endemic and all cases from elsewhere as sporadic. Rather, the geographic patterns describe a mosaic pattern with countries with high, intermediate, and low BL rates scattered in all continents, except in Asia, and North, South and Central America where none of the countries reached the threshold for high BL rate. This pattern suggests that there might be multiple risk factors that influence the within- and between-regional variation of BL rates. Because BL has a multi-factorial etiology, including EBV [63,64], *P falciparum* [65], HIV [25], and immune dysregulation, including from genetic or nutritional factors [66], the mosaic pattern observed here provides an epidemiological framework, as did the endemic/sporadic classification, for discovery of etiological factors that vary regionally, leading to pockets of high or deficit BL rates. Second, we confirm that BL predominates in males, including in countries that reported only pediatric cases. Finally, we show that BL risk is bimodal in all populations, and the proportions of pediatric versus adult-type BL vary geographically.

Our finding that BL rates were highest in Eastern (Malawi and Uganda) and Middle Africa (Cameroon) agrees with historical data [67]. This pattern is attributed to climatic co-factors favoring *P falciparum* transmission [68]. Exposure to holoendemic *P falciparum* for >6 months per annum [69,70] is the strongest geographic co-factor responsible for this pattern (Figure 1b). BL rates were intermediate or low in countries where climate is unfavorable to transmission of *P falciparum* [69,71]. Thus, the rate in Malawi was 11 times that in Kenya and 48 times that in Ethiopia, which lie at a high elevation (>5000 ft); 11–14 times those in Chad and Mali, which have a dry or arid climate (<20 inches of rainfall per year); and 11 times that in South Africa with a temperate climate (<16 °C) [70]. However, our finding that geographic patterns of BL rates vary across countries with apparently similar climate might be due to contribution of other etiological factors [72]. These factors could be related to micro-geographical variation in patterns of genetic complexity of *P falciparum* infection (number of different clones capable of causing an infection at the same time) [73–75] or parasites with distinct genetic profiles or strains [76,77]. Entomological factors, including type of mosquitoes, biting habits (night or day), host preference, and niche factors (where mosquitoes live) could also lead to subtle variation in BL rates [78]. Although not usually considered, other infections that are spread by mosquitoes could contribute [65]. Finally, the role of EBV variants [63,64], nutritional deficiencies, such as dietary selenium [79] or magnesium [66], are exposures that could modify geographical patterns of BL.

We confirm that BL from all regions is multimodal. The clearest evidence of multimodality was first reported in US data [46,47,58], but the results were not generalizable. Multimodality is an epidemiological manifestation of biologic heterogeneity [80,81]. Multimodal patterns were reported in a small series of HIV-negative BL cases from France that were diagnosed carefully [62], reducing concern that multimodality is due to diagnostic misclassification and increasing support for age-related biologic heterogeneity. Recent molecular studies of BL support the hypothesis of biological heterogeneity with age. A study of 162 well-diagnosed sporadic BL cases in Germany reported that the proportion of EBV + BL cases increased with the age at BL diagnosis [82]. Because EBV + BL is characterized by fewer somatic mutations in driver genes, particularly those in the apoptotic pathway, than EBV—BL [83], this finding suggests that EBV could be a biological marker of age-related heterogeneity in BL. This idea was also supported by findings that somatic mutations in *ID3*, *TCF3*, *CCND3*, and *SOX11* genes were detected more frequently in pediatric BL patients than adult cases [82], which agrees with the EBV data discussed above. Independent support for this idea has been reported in a study of 230 BL cases from Africa, the US, Germany, and Brazil, which showed distinct somatic mutations in pediatric cases (*ARID1A*) versus adult cases (*TET2*) [84]. A promising area of research is to elucidate the molecular basis of multimodality of BL in regional and global datasets.

We confirm the male predominance of BL, which was previously based on limited datasets [46,73,85], with worldwide BL data, including from countries with only pediatric cases. Sex patterns may suggest a role of reproductive factors or social factors related to gender, but these explanations are implausible because this pattern is observed in all countries with only pediatric or both pediatric and adult cases, and in countries with different socioeconomic statuses. The simplest explanation may be a role of recessive genetic factors on sex chromosomes [86]. Thus, females who carry one allele of a predisposing recessive genetic variant on the X-chromosome could experience lower risk of BL because the random lyonization of alleles could lead to escape from X-inactivation of the tumor-suppressor gene (abbreviated as EXITS) [87]. This idea is supported by recent findings of somatic mutations in *DDX3X* on the X-chromosome [88] and sex determining region Y-box transcription factor 11 (*SOX11*) in BL tumors [82]. However, germline studies are needed to confirm the population level effects of this hypothesis.

Our findings that BL rates in India and China are 1/20th those in Uganda and 1/6th the rate in US white people is puzzling given the role of *P falciparum* and EBV as risk factors for BL. Climate is suitable for *P falciparum* transmission in India (Figure 1b) [89]

and China, although transmission has recently been eliminated in China [90]. EBV infection is prevalent in Asia and its epidemiology is similar to that in Africa more than that in the US or Europe [91,92]. While underdiagnosis and/or underascertainment of BL are possible explanations for the discrepant low BL rates in Asia, that explanation is inconsistent with the high rates for nasopharyngeal cancer [93] and extra-nodal NK/T-cell lymphomas [94], both associated with EBV, in Asia. The low BL rates could be an epidemiological manifestation indicating that EBV strains circulating in Asia have a lower population attributable risk for BL than EBV in Africa [63]. EBV phylogenetic data have confirmed that EBV in Asia is genetically distinct from EBV in Africa [63,95–97]. Moreover, the EBV in US/Europe is phylogenetically closer to that in Africa [63,96], which is intriguing given that BL rates in US/Europe intermediate to those in Africa versus Asia. Molecular analysis of EBV variants in regions with different BL rates could lead to discovery of EBV variants with differential risk for BL.

Our findings that BL is multimodal raise several questions. First, could it be that adult cases are missed in those countries where only or mostly pediatric cases were reported? If so, why are adult cases missed? Second, the observation that compared to US white people, pediatric BL rates were twice as high in five European countries (Belgium, Netherlands, France, Italy, and Spain) and in two Asian countries (Israel and Turkey) was unexpected because both regions are thought to have a comparable risk profile for EBV and *P falciparum*. Similarly, the observation that adult BL rates were lower in most countries than rates among US white people is puzzling. It is difficult to explain why pediatric and adult BL rates vary. While it is theoretically possible that differences in case ascertainment, diagnosis, and registration of BL contribute, it is difficult to imagine how under-ascertainment could lead to different patterns for pediatric versus adult cases. As we noted in the Methods, BL might be coded differently. It was identified using ICD10 code C83.7, which is in the non-Hodgkin lymphoma range. BCL, a mature B-cell leukemia, Burkitt type that is assigned code C91.8 was, per IARC practice, not included with BL in CI5(XI). The US Surveillance, Epidemiology, and End Results (SEER) Program, includes the BCL ICD-0-3 morphology code 9826 with the BL morphology code 9687 to calculate BL rates, which increases US SEER BL cases by up to 15% [46,88]. Researchers need to be cognizant of these differences in handling of BL and BCL types when making comparisons in rates between our CI5 data and those reported using national registries or over time. However, this would not explain why pediatric versus adult rates would vary geographically as noted above.

The interpretation of BL rates assumes accurate and reliable diagnosis, classification, and registration [83,98–101]. However, BL data are clouded with uncertainty because a gold standard has not been established. The WHO has devoted efforts to improving and standardizing BL diagnosis through successive revisions of the Classification of Hematopoietic and Lymphoid Tissues, but the impact on global BL data is unknown. In 2008 [42], the WHO recommended that diagnosis of Burkitt-like lymphoma (BLL) be reclassified to B-cell lymphoma unclassifiable (BCL-U) with features intermediate between BL and diffuse large B-cell lymphoma (DLBCL). In 2016 [43], the WHO further recommended that cases still diagnosed as BLL based on having features like BL but with multiple chromosomal translocations, e.g., *MYC/BCL6*, or *BCL2*, be assigned a new name called high-grade B-cell lymphoma (HGBL) with double-hit or triple-hit lymphoma, or not otherwise specified (NOS). However, BL cases with features like BL that lacked *MYC* rearrangement but with 11q alterations (gain in 11q23.2-23.3 and losses of 11q24.1-ter) were assigned a new name BLL-11q. These cases, although technically not BL, are still coded along with BL morphology code 9687 in ICD-O3, highlighting the difficulties that still surround the definition, the accuracy, and reliability of BL data [43,102–104]. The 5th edition classification of haematolymphoid tumors leaves the definition of BL unchanged but stresses EBV as a marker of discrete biologic groups (EBV-positive versus EBV-negative BL) based on consistent findings of common molecular features regardless of epidemiologic context and

geographic location. Thus, EBV status should supersede epidemiological subtypes of endemic, sporadic, and immunodeficiency-associated [105]. While consensus guidelines should improve the reliability of diagnosis, accuracy is at the mercy of pathologists who diagnose cases and the quality of their experience, which is likely variable because BL is rare, and at the mercy of data abstractors and coders who translate technical diagnoses into classification codes that may be ambiguous [42,106,107]. The quality of these services likely varies between poorer [108] and richer areas based on better access to adequate tumor material, appropriate test equipment, reagents, and pathology referral support [41,109–111]. We performed a limited evaluation of NHL data (C82-86, C96) versus BL to assess to what extent quality of data (underascertainment or incomplete data) contributes versus variation in risk factors for five countries with most variable BL rates, namely, the US, Uganda, Hong Kong, China, and India. Focusing on NHL and BL rates in males, where number of BL cases are reasonably large, we observed that male NHL rates were highest in the US, for both white and black individuals, then in Uganda and Hong Kong, and then lower in other parts of China and India. The rates ranged from 1.8 to 7.6 per million in China, with a median of about 4.0. The rates ranged from 1.2 to 5.3 per million in India, with a median of about 3.0. The ratio of the BL to the NHL rate was 1.65 for Uganda, 0.34–0.43 in the US, and 0.14 in China and 0.23 in India. These patterns suggested to us that there is considerable geographic variation in both the BL and NHL rates, with the relative variation greater for BL than for NHL. We infer from this limited comparison that the differences in case identification and real differences in risk are playing a role in the observed patterns of BL. While issues of differences in case identification and registration might be greater in poorer countries, they may occur in richer countries based on whether patients access health care through private or single payer systems [112,113].

We believe uncertainty about the validity of BL diagnoses in different regions has reduced the enthusiasm to conduct BL research, particularly in poorer countries. Thus, almost all high-impact research on BL, such as using molecular methods to define BL [98–102], testing novel treatments [105], and developing? prognostic indices? [114–116] has been conducted in richer countries. The exclusion of poorer countries from BL research is ironic given that Denis Burkitt's epoch-making report was based on cases seen in Africa [12]. Burkitt's report is one of the most important epidemiological cancer discoveries of the last century, ranking close to the "two-hit hypothesis" by Alfred Knudson based on analysis of retinoblastoma data [117] and the hereditary cancer predisposition syndrome by Drs. Frederick Li and Joseph Fraumeni [118]. Those reports have inspired coordinated multidisciplinary global research [119] establishing retinoblastoma and the LFS as epidemiological models for discovery of genetic cancer predisposition and biology [120,121]. While the seminal discoveries in viral oncology, cancer chemotherapy, and characterization of the tumor lysis syndrome resulting from BL research showed its potential as a disease model for discovery [122], subsequent research has been less revealing because it is conducted in limited geographical and disciplinary settings [123], which limits the type of questions that can be answered. The unanswered questions include a reliable definition of the gold standard of BL diagnosis, BL subtypes, the multimodal patterns, discovery of EBV variants and predisposition to BL, development of simpler diagnostic methods, and better therapies. Our analysis shows that thousands of BL cases occur globally. These cases confirm the feasibility of conducting BL research globally to answer questions identified here. One mechanism could be modeled along the framework of the International Collaboration for Cancer Classification and Research (IC³R) under the World Health Organization. International research on BL could boost access to BL expertise in poorer countries, enable the recruitment of diverse BL cases for research to answer outstanding questions and support development of novel technology, such as liquid biopsy [124] and telepathology [111] to support BL research and care. International research on BL could deliver on the promise of BL envisioned six decades ago by Joseph Burchenal.

## 5. Conclusions

To summarize, we report the first epidemiological analysis of BL diagnosed worldwide. BL rates were highest in countries in Africa, intermediate in North America, Europe, and South/Central America, and lowest in Asia. BL rates varied within and between regions. BL rates in Asia, particularly India and China, were 1/20th of those in Africa. The deficit of rates in Asia could be due to underascertainment of cases in that region or could be an epidemiological clue that EBV variants circulating in Asia are associated with lower virulence for BL. We observed that BL predominates in males in all countries, including those with only pediatric cases. This pattern may be due to recessive genetic factors on sex chromosomes that influence the risk for BL. Our results show that BL is multimodal in all regions, providing epidemiological evidence for biologic heterogeneity of BL cases. Given the concerns about data quality discussed above, it will be important to repeat comprehensive epidemiological analyses of worldwide cases at reasonable intervals to reevaluate and update the patterns as the quality of the underlying data improves over time. The number of BL cases reported worldwide support the feasibility of conducting international transdisciplinary collaborative research on BL.

**Supplementary Materials:** The following supporting information can be downloaded at: https://www.mdpi.com/article/10.3390/hemato3030030/s1. Table S1: List of the CI5 (XI) registries and how the data were combined to estimate country-specific estimates. Table S2: Table showing BL counts and rates overall and by gender for all the countries whose data were downloaded from Cancer Incidence in Five Continents (XI) report IARC Scientific Publications No. 166 registries. Data are sorted by total case count within region. Only countries with ≥30 cases were included in the geographic analysis, those with ≥60 cases were included in the sex-specific analysis, while those with ≥100 cases were included in the age-specific analyses. Table S3: Table showing BL counts and rates overall and by gender for all the countries whose data were downloaded from the African Cancer Registry Network. Only countries with ≥30 cases were included in the geographic analysis, those with ≥60 cases were included in the sex-specific analysis, while those with ≥100 cases were included in the age-specific analyses. Table S4: Table showing age-specific counts, rates, and incidence rate ratios in CI5 (XI) and AFRCN countries with ≥100 cases

**Author Contributions:** S.M.M. conceived the study, guided the analysis, interpreted the data, and drafted the manuscript. S.S.D. analyzed the data, interpreted the data, and edited the paper. All authors have read and agreed to the published version of the manuscript.

**Funding:** This research was funded by the Intramural Research Program of the Division of Cancer Epidemiology and Genetics, National Cancer Institute, National Institutes of Health, Department of Health, and Human Services Contracts HHSN261201100063C and HHSN261201100007I.

**Institutional Review Board Statement:** Ethical review and approval were not applicable because we analyzed anonymized data available via public datasets.

**Informed Consent Statement:** Not applicable for studies not involving humans.

**Data Availability Statement:** The datasets used and/or analyzed during the current study can be downloaded from the CI5-XI website (https://ci5.iarc.fr/CI5-XI/Default.aspx; accessed on 21 August 2021) using the Help/Download option or requested from AFCRN secretariat (https://afcrn.org/index.php/research/researches-andcollaborations; accessed on 22 October 2021).

**Acknowledgments:** We thank Jacques Ferlay of the International Agency for Research on Cancer for granting access to data files. We thank Maxwell D. Parkin, Biying Liu, and all of the registries, members of the African Cancer Registry Network (AFRCN) (http://afcrn.org/index.php/membership/membership-list; accessed on 22 October 2021), for giving us their BL data from Africa for 2018. We thank Marianne Hyer, Emily Carver, and Jeremy Lyman of Information Management Systems (Rockville, Maryland) for preparing the files for analysis, drawing the maps and bar charts, and David Check of the Division of Cancer Epidemiology and Genetics, National Cancer Institute (Bethesda, Maryland) for drawing age-specific graphs and incidence rate ratio graphs as well as polishing other figures. This article is dedicated to Elaine S. Jaffe who has dedicated her service to using pathology to help the definition and discovery in lymphoma.



**Conflicts of Interest:** The authors declare no conflict of interest. The funders had no role in the design of the study; in the collection, analyses, or interpretation of data; in the writing of the manuscript, or in the decision to publish the results.

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
