# Peer review of "Burkitt Lymphoma Incidence in Five Continents"

_hemato, doi:10.3390/hemato3030030_

Round 1

Reviewer 1 Report

Major comments:

Overall this is a strong and important manuscript with a very well written intro and discussion that provides proper nuance to the data. When discussing the limitations of the data and in the conclusion itself I think it's important to note that this analysis should be continuously reevaluated as the quality of the underlying data improves over time.

Only major revision I would request would be around the following. I wonder if there isn't a way to use the data you have access to to improve the manuscript a bit. Is there a way to look at the distribution of the NHL diagnoses across countries to get an idea of the granularity of NHL diagnoses made in each country. Perhaps some analysis of the granularity of the ICD codes by country will allow you to account for some of the unexpected findings in a supplementary table? I certainly wonder if some of the unexpected findings in India and China for example are due to limitations in pathology capacity--or maybe not capacity but limitations in the clinical utility or need for more granularity in diagnosis leads to stopping at a higher level diagnosis without specifically narrowing in on Burkitt lymphoma. Likewise, some of the underlying data from the African Cancer Registry is limited in its reporting clearly. I am certain that there are adult cases of Burkitt lymphoma in some of the sub-Saharan African countries so I wonder if there is some way to demonstrate the quality of data across the country--is the data on ages missing and limited or is the coverage of the population limited? Or are all diagnoses made by FNA only without capacity for tissue histology?

Minor comments:

Line 110-112 are hard to understand. Please rewrite in another way.

Author Response

Comment: Overall this is a strong and important manuscript with a very well written intro and discussion that provides proper nuance to the data. When discussing the limitations of the data and in the conclusion itself I think it's important to note that this analysis should be continuously reevaluated as the quality of the underlying data improves over time.

 Response: We thank the reviewer for this suggestion. We have revised the manuscript at  lines 463-466 as follows: “Our results show that BL is multimodal in all regions, providing epidemiological evidence for biologic heterogeneity of BL diagnose in all regions. Given the concerns about data quality discussed above, it will be important to repeat comprehensive epidemiological analyses at reasonable intervals  to reevaluate and update the patterns as the quality of the underlying data improves over time.

Comment: Only major revision I would request would be around the following. I wonder if there isn't a way to use the data you have access to improve the manuscript a bit. Is there a way to look at the distribution of the NHL diagnoses across countries to get an idea of the granularity of NHL diagnoses made in each country? Perhaps some analysis of the granularity of the ICD codes by country will allow you to account for some of the unexpected findings in a supplementary table? I certainly wonder if some of the unexpected findings in India and China for example are due to limitations in pathology capacity--or maybe not capacity but limitations in the clinical utility or need for more granularity in diagnosis leads to stopping at a higher-level diagnosis without specifically narrowing in on Burkitt lymphoma. Likewise, some of the underlying data from the African Cancer Registry is limited in its reporting clearly. I am certain that there are adult cases of Burkitt lymphoma in some of the sub-Saharan African countries, so I wonder if there is some way to demonstrate the quality of data across the country—

Response: We thank the review for these excellent questions and suggestions. While time constraints to complete this project and the complexity of integrating the new results in the manuscript make a full exploration difficult, we have done a limited evaluation of the NHL data to compare patterns of rates of NHL overall versus BL and also compared the BL/NHL rate ratios for the US, Africa (Uganda) and Asia (China and India) as illustrative examples. This limited analysis shows that , it looks like there is considerable geographic variation in both the BL and NHL rates, with the relative variation greater for BL than for NHL. These patterns support the conclusion that both differences in case identification and real differences in risk are playing a role. We have revised the discussion as on lines 404-418 follows:

“We performed a limited evaluation of NHL data (C82-86, C96) versus BL to assess to what extent quality of data (under-ascertainment or incomplete data) contributes versus variation in risk factors for five countries with most variable rates, namely, US, Uganda, Hong Kong, China and India.  Focusing on rates in males, where number of BL cases are larger, we observed that male NHL rates were highest in the US, both white and black individuals, then in Uganda, Hong Kong, and then lower in other parts of China and India.  The rates ranged from 1.8 to 7.6 per million in China, with a median of about 4.0.  The rates ranged from 1.2 to 5.3 per million in India, with a median of about 3.0.  The ratio of the BL to the NHL rate was 1.65 for Uganda, 0.34-0.43 in the US, and 0.14 in China and 0.23 in India.  These patterns suggested to us that there is considerable geographic variation in both the BL and NHL rates, with the relative variation greater for BL than for NHL.  We infer from this limited comparison that the differences in case identification and real differences in risk are playing a role to the observed patterns. While issues of differences in case identification and registration might be greater I poorer countries, they…”.

We also agree with the reviewer’s concern that limitations in pathology capacity, in clinical utility, the limited granularity in diagnosis, which may lead to stopping diagnosis formulation at a higher-level diagnosis without specifically narrowing in on Burkitt lymphoma, could all affect the comparability of population results. Likewise, the paucity of adult cases of Burkitt lymphoma in sub-Saharan African countries is noted. Thus, we strongly advocate for a research effort focused on BL to identify these issues and generate solutions that would help improve the quality of BL data within each region and globally because use of routine data repositories may to provide an accurate picture of BL rates in populations.

Comment: is the data on ages missing and limited or is the coverage of the population limited?

Response: The data we accessed was mostly complete for age of the cases and the source population. However, it is possible that these problems could affect BL data differentially, especially in Africa, where most cases are below 14 years. Therefore, it is not possible to draw reliable conclusions about whether most cases in Africa are indeed pediatric  whereas cases elsewhere are adult.

Comment: Or are all diagnoses made by FNA only without capacity for tissue histology?

Response: We agree with the reviewer that diagnosis by FNA could be part of the problem resulting in higher level diagnosis of NHL, but not BL. This problem is likely to be greater in Asia and Africa. We advocate for multi-regional collaboration to identify and resolve these issues.

Minor comments:

Comment: Line 110-112 are hard to understand. Please rewrite in another way.

Response: We have revised this as follows “To minimize the impact of sparse data on the rates, we restricted analyses to countries with a minimum of total cases. Specifically, 30 total cases for geographic analyses of overall rates (38 countries), 60 total cases for sex-specific analyses (28 countries), and 100 total cases for age-specific analyses (18 countries), referred to as the 30/60/100 criteria.”

Reviewer 2 Report

The work is well written and methodologically correct as far as the sources of data acquisition are correct.

Unfortunately, the network of cancer registries is not uniformly distributed in different parts of the world. The limits of the work are related to the limits of the data sources used.

The Materials and Methods are described in full and clear form.

The Discussion is correctly articulated; the role of EBV and the mention of genetic aspects in favoring higher rates in the male gender is very interesting. The conclusions are clearly described.

Author Response

The work is well written and methodologically correct as far as the sources of data acquisition are correct.

Comment: Unfortunately, the network of cancer registries is not uniformly distributed in different parts of the world. The limits of the work are related to the limits of the data sources used.

Response: We agree with the reviewer that the coverage of cancer registries is an issue, which is part of the data. 

The Materials and Methods are described in full and clear form.

The Discussion is correctly articulated; the role of EBV and the mention of genetic aspects in favoring higher rates in the male gender is very interesting. The conclusions are clearly described.

Round 2

Reviewer 1 Report

I appreciate the revisions and deep thinking about the suggestions. It's a very difficult topic to study but with the proper nuance, as provided by the authors, it is very much worth sharing and to continue to dive into with deeper granularity over time.